# Noise Pruning in Neural Radiance Fields via Influence Functions and Segmentation

## Abstract

Neural Radiance Fields (NeRF) is a method for 3D scene modeling that employs fully-connected networks to learn 3D geometric information and synthesizes high-quality novel views. However, NeRF exhibits vulnerability when confronted with distractors in the training images, such as the presence of moving objects like pedestrians or different weather conditions within specific views. Given the difficulty of data curation in NeRF compared to other domains, training a robust model that maintains 3D consistency is an important and timely challenge. Previous approaches have attempted to differentiate distractors by using loss values, but there is a fundamental limitation that hard-to-learn pixels like high-frequency details also show high loss values. In this paper, we propose a noise pruning framework via influence functions to effectively filter out noisy pixels, ultimately enhancing the robustness of NeRF. Furthermore, we improve the precision of detection by incorporating segmentation techniques to refine pixel-level predictions. Our method demonstrates superior performance on benchmark datasets, including synthetic and natural scenes, showcasing its effectiveness across various environments and proficiency in dataset pruning.

## 1 Introduction

Recent advances in deep learning have exhibited remarkable capabilities across various domains, including natural language processing (Devlin et al., 2018; Vaswani et al., 2017; Brown et al., 2020; Liu et al., 2019) and computer vision (Simonyan & Zisserman, 2014; He et al., 2016; Redmon et al., 2016). They also have notably excelled in 2D image processing and, as widely known, found diverse applications. This surge in deep learning's success has fueled a demand for extending its capabilities into the realm of 3D vision technology, which is pivotal in applications like virtual reality (Sherman & Craig, 2003; Anthes et al., 2016; Wohlgenannt et al., 2020), autonomous driving (Bojarski et al., 2016; Kiran et al., 2021; Sun et al., 2020), and healthcare (Ronneberger et al., 2015; Hatamizadeh et al., 2021; Balakrishnan et al., 2019).

In response to this growing demand, extensive research is underway on techniques based on point clouds (Qi et al., 2017a;b; Zhao et al., 2021), voxels (Qi et al., 2016; Choy et al., 2019), and polygons (Su et al., 2015; Kokkinos et al., 2012) to enable the learning of 3D objects or scenes. However, most of these methods suffer from reliance on 3D supervision, which is a costly task and often elusive requirement in typical scenarios. To address the limitations associated with the need for 3D supervision, a novel approach known as the Neural Radiance field (NeRF) (Mildenhall et al., 2021) has been proposed. NeRF is an advanced 3D scene representation method that leverages fully connected networks to model both the appearance and geometry of objects and scenes from a partial set of 2D images. It takes 3D spatial coordinates and viewing directions as inputs and generates RGB color and volume density as outputs. By tracing camera rays through the scene and utilizing the RGB and density information of points along those rays, it is capable of rendering scenes. As a result, NeRF takes advantage of 2D supervision to synthesize high-quality novel views, offering a promising pathway to bridging the gap between 2D and 3D vision capabilities.

Nevertheless, NeRF does have a limitation—it necessitates well-curated 2D images devoid of noisy artifacts to learn proper 3D consistency during training. If individual 2D images of ground truth

offer conflicting guidance to the model due to distracting factors like the presence of a pedestrian walking through a specific view, the integrity of 3D consistency may easily deteriorate.

A straightforward solution is a careful dataset curation to prevent the inclusion of such noise. However, considering the inherent difficulty of dataset construction in NeRF domain, additional burdens to ensure the devoid of real-world factors, such as moving objects, weather shifts, fluctuating lighting conditions, and varying capture times, are challenging. Due to these factors, NeRF has a relatively limited number of benchmark datasets compared to other computer vision tasks like point cloud processing and 2D vision.

Another approach aims to pursue a robust model capable of withstanding such challenges. NeRF-W (Martin-Brualla et al., 2021) has ventured into enabling training on unstructured image collection from the internet by incorporating the concept of uncertainty to distinguish static from transient objects, but they exhibit performance limitations. Similarly, some studies leverage the time axis to separate static and dynamic components within a scene (Wu et al., 2022; Li et al., 2021; Gao et al., 2021). Though effective, they pose a significant challenge as the training images must be arranged chronologically, like a video. In addition, semantic segmentation models can be utilized to preemptively identify and exclude undesirable objects within training views (Tancik et al., 2022). While this approach effectively handles pre-trained object classes, such as pedestrians and vehicles, it performs less reliably when dealing with unfamiliar objects.

More recently, RobustNeRF (Sabour et al., 2023) introduced a framework that generates masks for identifying outlier objects based on pixel losses and integrates them as weights in iteratively reweighted least squares(IRLS). Nevertheless, due to its reliance on pixel losses, inherent limitations arise when attempting to discern the high-frequency details of inliers, which are difficult to learn, from outliers based on the loss magnitude. Additionally, the proposed masking technique is based on the assumption that outliers occupy large and connected regions of an image. This assumption can result in notable performance degradation when dealing with other types of noise rather than large objects.

Given this context, training a robust model despite the presence of noisy training data is a highly significant yet challenging task. To address this issue, we propose an effective noise pruning framework that leverages influence functions, a mathematical technique for quantifying the impact of individual data points on the model's predictions. Inspired by the idea that noisy pixels, including moving objects or weather-related changes, can be viewed as mislabeled data points, we identify and exclude pixels that disrupt 3D consistency across other views by calculating sample-wise influence scores.

Our framework excels in synthesizing high-quality novel views from noisy datasets while simultaneously refining the dataset by removing noise pixels using influence scores. Furthermore, we improve the precision of the influence function by incorporating segmentation techniques to refine pixel-level noise candidates. Considering the inherent challenges in dataset curation within the NeRF domain, we believe that our framework unlocks the potential of constructing large-scale datasets comprising diverse scenes by effectively pruning the noisy pixels.

Our contributions are summarized as follows:

- We first pioneer the integration of influence functions into NeRF, effectively excluding distractors that disrupt the 3D consistency. Also, we demonstrate the improved performance of novel view synthesis on noisy datasets.
- We incorporate panoptic segmentation techniques into our framework to enhance the precision of pixel-level noise candidates results from influence functions. We employ SAM (Kirillov et al., 2023), which has recently showcased remarkable zero-shot performance. Our framework can be seamlessly combined with any segmentation model capable of partitioning the scene into small regions.
- We first propose dataset pruning in NeRF, removing distracting pixels from noisy datasets. This process enables easier data construction, bringing out the potential of constructing large-scale datasets for NeRF, similar to those in 2D vision.

## 2 PRELIMINARIES

### 2.1 NEURAL RADIANCE FIELD

Neural Radiance Field (NeRF) (Mildenhall et al., 2021) represents a 3D scene as a continuous function $f_\theta$, implemented as a multi-layer perceptron (MLP) network. It estimates RGB color $c$ and volume density $\sigma$ from 3D coordinates $\mathbf{x} = (x, y, z)$ and 2D viewing direction $\mathbf{d} = (\theta, \phi)$ as $\{\mathbf{c}, \sigma\} = f_\theta(\mathbf{x}, \mathbf{d})$. In pursuit of 3D-consistent geometry, volume density $\sigma$ depends solely on the coordinates $\mathbf{x}$ in order to remain constant regardless of changes in viewing direction $\mathbf{d}$. In contrast, RGB color $\mathbf{c}$ additionally incorporates viewing direction $\mathbf{d}$ to represent non-Lambertian surfaces.

Given a camera ray $\mathbf{r}(t) = \mathbf{o} + t\mathbf{d}$, the corresponding pixel color $\hat{C}(\mathbf{r})$ is approximated by integrating the radiance within the interval from the camera's near plane $t_n$ to the far plane $t_f$:

$$\hat{C}(\mathbf{r}) = \int_{t_n}^{t_f} T(t)\sigma(\mathbf{r}(t))\mathbf{c}(\mathbf{r}(t), \mathbf{d})dt, \text{ where } T(t) = \exp\Big(-\int_{t_n}^{t} \sigma(\mathbf{r}(s))ds\Big) \tag{1}$$

where $\mathbf{o}$ and $T(t)$ denote the camera center and the accumulated transmittance, respectively.

Then, NeRF is optimized using a photometric loss with the ground truth pixel colors $C(\mathbf{r})$ as follows:

$$\mathcal{L} := \sum_{\mathbf{r} \in \mathcal{R}} \left\| C(\mathbf{r}) - \hat{C}(\mathbf{r}) \right\|_2^2 \tag{2}$$

where $\mathcal{R}$ is the set of rays. It is important to mention that NeRF typically assumes a well-curated training dataset: thus, the presence of noise in the training dataset significantly degrades performance. More specifically, if individual ground truth pixel colors provide conflicting guidance to the model due to the existence of distractors, it becomes challenging to learn consistent geometry across the views (Martin-Brualla et al., 2021; Sabour et al., 2023).

### 2.2 INFLUENCE FUNCTIONS

Influence function is a mathematical technique originating from classical robust statistics to measure the impact of individual data points in the trainset on a model's predictions (Hampel, 1974; Koh & Liang, 2017). Leave-one-out (LOO) retraining is a straightforward approach to estimating the influence of a data point by excluding it from the trainset, retraining the model, and then assessing the change in the model's prediction. However, conducting LOO retraining for every data point is computationally intractable. To mitigate the prohibitive cost, influence functions are proposed as approximate versions of LOO.

In empirical risk minimization (ERM), we minimize the finite-sum objective for a training dataset $D := \{z_i : (x_i, y_i)\}_{i=1}^N$ as

$$\theta^* := \arg\min_\theta \mathcal{L}(D, \theta) \tag{3}$$

where $\theta$ is the network parameter and $\mathcal{L}$ is the sum of sample-wise loss over the dataset: $\mathcal{L}(D, \theta) = \frac{1}{N} \sum_{i=1}^N \ell(z_i, \theta)$. Here, the estimated impact of a data point $z_i$ on another sample $z_j$ using LOO retraining is defined as

$$\mathcal{I}_{LOO}(z_i, z_j) := \ell(z_j, \theta^*_{-z_i}) - \ell(z_j, \theta^*) \tag{4}$$

where $\theta^*_{-z}$ is retrained parameter without $z$. To mitigate prohibitive computational costs, influence functions are proposed as approximated versions of LOO retraining (Koh & Liang, 2017) as

$$\mathcal{I}_\epsilon(z_i, z_j) := \ell(z_j, \theta^*_{-z_i, \epsilon}) - \ell(z_j, \theta^*). \tag{5}$$

where $\epsilon$ is the up-weighting parameter on $z_i$, and the corresponding up-weighted loss of $z$ is defined as

$$\theta^*_{-z_i, \epsilon} = \arg\min_\theta \Big(\mathcal{L}(D, \theta) + \epsilon \cdot \ell(z_i, \theta)\Big).$$

Then, Influence Function is represented as

$$\mathcal{I}(z_i, z_j) := \frac{d\mathcal{I}_\epsilon(z_i, z_j)}{d\epsilon}\Big|_{\epsilon=0} = \nabla_\theta \ell(z_j, \theta^*)^\top H^{-1} \nabla_\theta \ell(z_i, \theta^*)$$

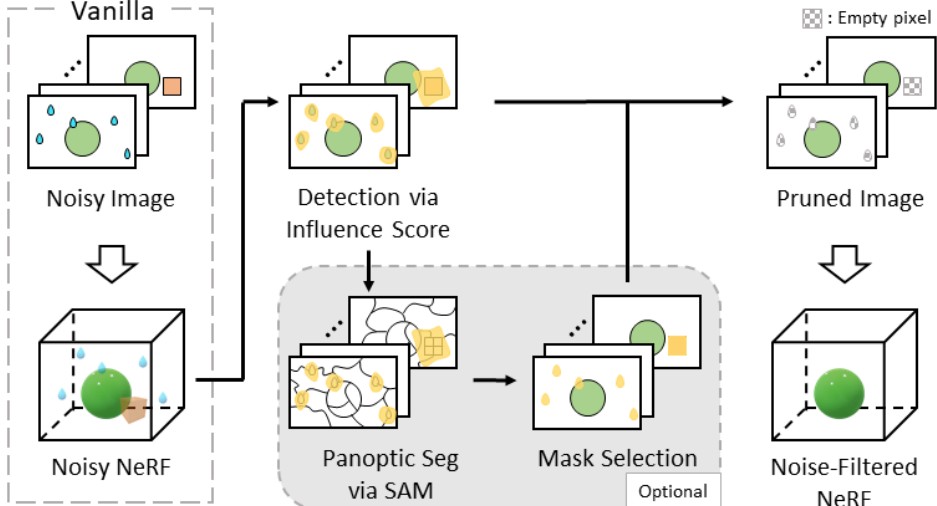

Figure 1: **Overview of our framework.** In the Vanilla approach, certain regions become blurry due to the presence of distractors. We employ influence function to detect distractors. We obtain panoptic masks usin SAM and perform mask selection based on the influence scores to clearly segment distractors. Finally, through pixel pruning, we generate a clear image. This approach enables us to train a noise-filtered NeRF model.

where $H := \nabla_\theta^2 \mathcal{L}(T, \theta^*) \in \mathbb{R}^{P \times P}$ is the Hessian of the loss function with respect to the parameter $\theta^*$.

In typical scenarios, $z_j$ is assigned to the validation set when it is feasible. However, in a situation where this is not possible, $z_j$ is equated with $z_i$ itself. This enables us to compute self-influence, which signifies how well the model, trained on the remaining data points in the absence of $z_i$, can predict on $z_i$. A higher score indicates more difficult to predict, implying that the data point deviates from the majority of the trainset.

## 2.3 SEGMENTATION

Image segmentation is a crucial part of visual understanding, which involves dividing images into distinct parts or objects. Among various segmentation tasks, panoptic segmentation (Kirillov et al., 2019) stands out as it unifies both semantic and instance segmentation (Long et al., 2015; He et al., 2017). Panoptic segmentation assigns a unique label to each pixel in an image $I$, providing comprehensive information about the object's category $c$ and its individual instance $i$, as

$$\mathcal{S}(I) = \left\{ M_{c,i} \mid \cup_{c,i} M_{c,i} = I, M_{c,i} \cap M_{c',i'} = \emptyset \; \forall i \neq i' \right\} \tag{6}$$

where $\mathcal{S}$ is a segmentation model, and $M$ is segmentation mask.

SAM (Kirillov et al., 2023), a recent foundational segmentation model, achieve impressive zero-shot performance by utilizing an extensive segmentation dataset and generating finely detailed masks due to a high masks-per-image ratio.

## 3 METHOD

In this section, we introduce an effective noise pruning framework leveraging influence functions to identify noisy pixels, approximating leave-one-out (LOO) retraining to alleviate the prohibitive costs. First, we clarify the rationale behind opting for influence functions instead of loss-based approaches, with a particular focus on their relevance within the NeRF domain. Additionally, we incorporate a segmentation model into our framework to improve detection precision. Lastly, we delve into the importance of dataset pruning in NeRF and its potential for constructing large-scale datasets. The framework is visualized in Figure 3.

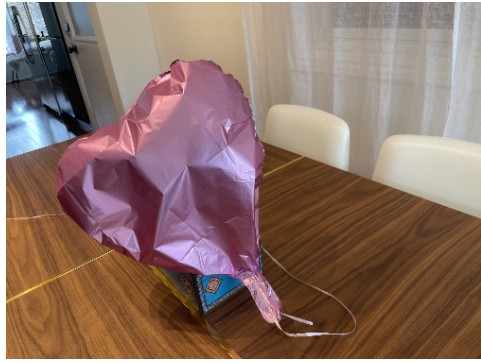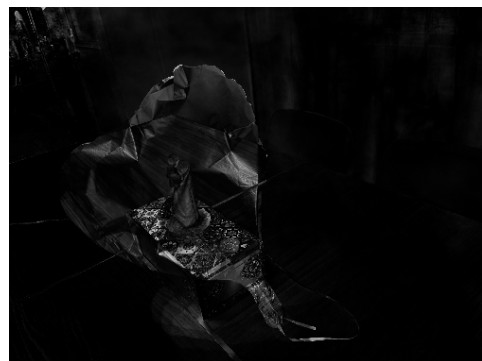

Figure 2: **Natural scenes.** (Left) Examples of training frames in Statue. (Right) Pixel losses of the corresponding frame, with brighter indicating higher loss. Some regions on the distractor exhibit high loss values due to the hard-to-learn object that lies beyond the distractor.

**Detection via Influence Function.**    In the training phase of NeRF, similar to other deep learning tasks, we minimize the finite-sum objective function for the pixels in the training dataset, as defined in equation 2. Throughout this procedure, the model learns to ensure 3D consistency across the ground truth 2D views captured from diverse directions within a single 3D scene. Therefore, even if typical pixels are omitted from the training dataset, the model can still predict them well by leveraging information from other data points. In contrast, in the case of noisy pixels, excluding them results in a notable decline in the model's performance for those pixels due to their inconsistent 3D geometry with the major stream of trainset.

In this context, pixel loss is commonly employed to differentiate them (Sabour et al., 2023); however, given that both distractors and high-frequency details can often yield high loss values, this poses a challenge for accurate differentiation. Furthermore, there may be underlying other factors contributing to the high loss values for distractors. We observed that it is a common phenomenon that when distractors exist only in specific views, relative 3D regions are recognized as empty spaces for most views. Therefore, as shown in Figure 2, even if ground truth distracting pixels exhibit high loss values, it could be due to the hard-to-learn object beyond the distractors.

Under these circumstances, it is not far-fetched to explore alternative approaches. Motivated by the concept that pixels associated with distractors can be considered mislabeled data points, we employ influence functions on the pre-trained model to identify pixels that deviate from the overall dataset consistency. We calculate the self-influence score $I(\mathbf{r}, \mathbf{r})$ per sample for $\mathbf{r} \in \mathcal{R}$ as in equation 6 and consider the top-ranked pixels group $G_{top}$ as noise pixel candidates.

**Refinement using Segmentation Model.**    While influence functions demonstrate noteworthy performance, we incorporate the segmentation method into our framework to further improve prediction accuracy. Specifically, we integrate panoptic segmentation to refine outputs from influence functions. It is important to note that segmentation need not be pre-trained on specific classes of objects; its capability to partition the scene into small parts is sufficient. To this end, we leverage SAM (Kirillov et al., 2023), which has recently shown impressive zero-shot performance in panoptic segmentation tasks. Specifically, we first utilize SAM to generate distinct masks for each individual parts of a scene $\{M_{c,i}\}$. Next, as mentioned earlier, we compute self-influence for each pixel and select the top-rank $G_{top}$ group based on the magnitude of self-influence. Not only that, now we also choose the bottom-rank $G_{bottom}$ group with the smallest self-influences. To determine whether to exclude each part segment, we asses how many pixels within the segment belong to either $G_{top}$ or $G_{bottom}$ groups. In other words, the pixel within the segment vote on whether the segment to which they belong is a distractor or not. Here, pixels that do not belong to either $G_{top}$ or $G_{bottom}$ do not participate in the voting. Finally, if the pixels within the segment belong more to $G_{top}$ than $G_{bottom}$, we prune the entire segment. The sizes of $G_{top}$ and $G_{bottom}$ are hyperparameters that impact the overall performance of our framework. In all our experiments, we do not perform separate tuning for them. Instead, to be as conservative as possible in pruning outliers, we fix $G_{top}$ representing outliers as the only $1\%$ of the entire data points and $G_{bottom}$, the clearly identified inlier group, as $30\%$. In most cases of our experiments, this refinement process with the segmentation model presents more accurate and object-centric results.

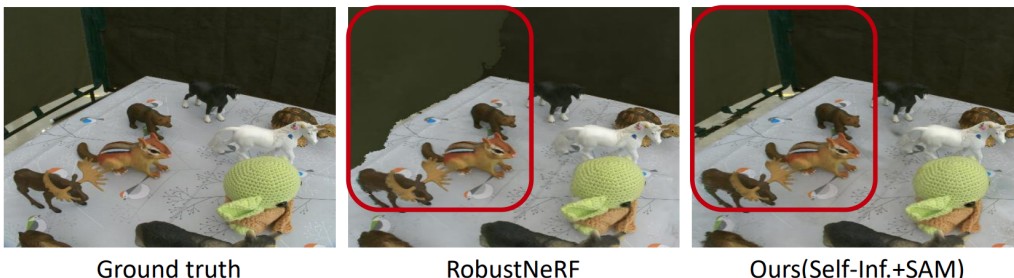

Ground truth        RobustNeRF        Ours(Self-Inf.+SAM)

Figure 3: **Qualitative evaluation.** As shown in the red box, RobustNeRF (Middle) fails to preserve the static components in Yoda scene. However, our approach (Right) can effectively learn intricate details within the scene without interference from distractors.

**Dataset Prunining.** Dataset construction in NeRf is known to be challenging due to a multitude of constraints compared to other domains. Subsequently, there are relatively few publicly available benchmarks. In this context, our framework offers significant merits by enabling the refinement of datasets even when they contain some noise. Additionally, being a pruning method, it allows for the refinement and aggregation of these raw datasets, which leads us to believe in the potential for large-scale dataset construction, similar to what has been achieved in the 2D vision domain.

## 4 EXPERIMENTS

### 4.1 EXPERIMENTAL SETTINGS

We use Mip-NeRF 360 (Barron et al., 2022) without any modifications to its architecture, following the previous work (Sabour et al., 2023): Proposal MLP with 4 layers and 256 hidden units, NeRF MLP with 8 layers and 1024 hidden units. The model is trained for 250k iterations with a batch size of 16,384. We empoloy Adam optimizer (Kingma & Ba, 2014) with hyperparameters $\beta_1 = 0.9$, $\beta_2 = 0.999$, and $\epsilon = 10^{-6}$. The initial learning rate is set at $2 \times 10^{-3}$ and exponentially decayed to $2 \times 10^{-6}$. The first 512 iterations are used for warm-up. To enhance training stability, we utilize Charbonnier loss (Charbonnier et al., 1994) instead of the $l_2$ loss (Krogh & Hertz, 1991). For evaluation, we report PSNR (Dosselmann & Yang, 2005), MS-SSIM (Wang et al., 2003), and SSIM (Wang et al., 2004) as performance metrics.

**Datasets** We employ both synthetic and real-world scenes to ensure the comprehensive evaluation of our methodology. The synthetic dataset was obtained from $D^2$NeRF (Wu et al., 2022), comprising 5 different scenes: *Car, Cars, Bag, Chairs* and *Pillow*. These scenes were generated using the Kubric dataset generator (Greff et al., 2022). Each scene has a 200-frame video sequence for training and 100 novel views for validation. Each training frame may contain distractors, whereas the validation images exclusively contain static components. Moreover, they possess ground truth labels for distractors in each training frame, facilitating a quantitative evaluation of our data pruning methodology.

For the natural dataset, we utilized publicly accessible benchmark datasets from Robust-NeRF (Sabour et al., 2023): *Statue, Android* and *BabyYoda*. In each of these scenes, distractors are distributed across extensive and interconnected regions, exhibiting variations in both their positions and types across frames. A detailed explanation for datasets is suggested in Appendix A.1.

We conducted comprehensive comparisons of our framework with baselines, including MipNeRF 360 (Barron et al., 2022) with $l_2$ loss and Charbonnier loss by following the previous work (Sabour et al., 2023). Furthermore, we evaluated our method against RobustNeRF, known for its robustness in scenarios where distractors occupy large and connected regions. While both approaches exclude distractors during training, there is a difference in the timing of their exclusion. Our method prunes noisy samples at the beginning of the training phase. However, RobustNeRF takes a gradual refinement approach, excluding distractors progressively as training iterations advance. To comprehensively assess the model's performance, we further compare the results with the upper bound, which

Table 1: Quantitative evaluation on the synthetic dataset. please note that 'SI' represents self-influence and 'SI+Seg.' denotes the whole framework combining self-influence with segmentation. Our data pruning approach results in a PSNR gain of minimum 2.5db to maximum 11dB compared to baselines.

| Method | Car | | Cars | | Bag | | Chairs | | Pillow | |
|---|---|---|---|---|---|---|---|---|---|---|
| | MS-SSIM ↑ | PSNR ↑ | MS-SSIM ↑ | PSNR ↑ | MS-SSIM ↑ | PSNR ↑ | MS-SSIM ↑ | PSNR ↑ | MS-SSIM ↑ | PSNR ↑ |
| NeRF-W | .814 | 24.23 | .873 | 24.51 | .791 | 20.65 | .681 | 23.77 | .935 | 28.24 |
| NSFF | .806 | 24.90 | .376 | 10.29 | .892 | 25.62 | .284 | 12.82 | .343 | 4.55 |
| NeuralDiff | .952 | 31.89 | .921 | 25.93 | .910 | 29.02 | .722 | 24.42 | .652 | 20.09 |
| $D^2$NeRF | .975 | 34.27 | .953 | 26.27 | .979 | 34.14 | .707 | 24.63 | .979 | 36.58 |
| mip-NeRF 360 | .911 | 27.58 | .920 | 24.82 | .982 | 41.02 | .970 | 36.00 | .912 | 33.12 |
| Ours (SI) | .976 | 37.37 | .945 | 26.67 | .990 | 42.52 | .975 | 36.64 | .943 | 35.13 |
| Ours (SI+Seg.) | .976 | 37.41 | .953 | 27.16 | .988 | 42.34 | .978 | 37.22 | .978 | 36.86 |
| Clean | .992 | 40.62 | .977 | 27.16 | .996 | 42.72 | .989 | 38.72 | .990 | 37.99 |

Table 2: Evaluation on noise detection.

| Method | Car | | Cars | | Bag | | Chairs | | Pillow | |
|---|---|---|---|---|---|---|---|---|---|---|
| | Acc. | F1 | Acc. | F1 | Acc. | F1 | Acc. | F1 | Acc. | F1 |
| Ours (SI Top-g.t.) | .83 | .81 | .85 | .83 | .76 | .74 | .78 | .76 | .72 | .69 |
| Ours (SI+Seg.) | .95 | .83 | .96 | .86 | .66 | .70 | .98 | .72 | .88 | .78 |

Table 3: Ablation on sensitivity.

| Method | Car |
|---|---|
| | PSNR ↑ |
| Ours (Self-Inf.2) | 32.92 |
| Ours (Self-Inf.5) | 37.37 |
| Ours (Self-Inf.10) | 37.12 |

are obtained by training Mip-NeRF 360 with clean data, which does not involve any distractors. A more detailed explanation of baselines is suggested in Appendix A.2.

## 4.2 RESULTS IN SYNTHETIC SCENES

Upon evaluating our approach in synthetic scenes, our method outperforms baseline methods in both PSNR and MS-SSIM metrics, as shown in Table 1. In fact, for the *Cars* and *Bag* scenes, our method shows performance close to the oracle. Since baseline models utilize conflicting guidance from the ground truth, the models could be confused to represent 3D-consistent scenes. However, since our method ignores distractors right from the beginning of the training phase, individual ground truth pixel colors do not provide conflicting guidance to the model, allowing it to effectively learn 3D-consistent geometry. Table 2 illustrates that our method can prune only the parts corresponding to distractors with high accuracy.

## 4.3 RESULTS IN NATURAL SCENES

Our method performs on par or even better than baselines, including RobustNeRF, as shown in Table 4. As depicted in Figure 3, it is evident that RobustNeRF struggles to capture scene details and static components effectively. In contrast, our approach adeptly excludes distractors while retaining the integrity of other static components during training.This discrepancy arises from the fact that RobustNeRF masks out regions where the loss magnitude is high across large, interconnected areas. This over-smoothing of the mask leads to the exclusion of areas that should not be omitted during training. Conversely, our method leverages influence functions without making any assumptions about distractors, allowing us to accurately identify pixels that adversely affect the training process and effectively detect distractors. Consequently, our framework minimizes the impact of distractors while preserving other essential details.

Table 4: Quantitative evaluation on the natural dataset.

| Method | Statue | | Android | | BabyYoda | |
|--------|--------|--------|---------|--------|----------|------|
| | SSIM ↑ | PSNR ↑ | SSIM ↑ | PSNR ↑ | SSIM ↑ | PSNR |
| mip-NeRF 360 ($L_2$) | .66 | 19.09 | .75 | 19.37 | .75 | 22.97 |
| mip-NeRF 360 (Ch.) | .73 | 19.64 | .66 | 19.53 | .80 | 25.22 |
| $D^2$NeRF | .49 | 19.09 | .57 | 20.61 | .65 | 17.32 |
| RobustNeRF | .74 | 21.05 | .72 | 22.78 | .81 | 29.95 |
| Ours (Self-Inf.) | .70 | 20.43 | .70 | 21.96 | .76 | 29.21 |
| Ours (Self-Inf. + SAM) | .77 | 21.52 | .71 | 22.45 | .83 | 30.5 |
| MipNeRF360 (Clean) | .80 | 23.57 | .71 | 23.10 | .84 | 32.63 |

### 4.4 PERFORMANCE ON DATASET PRUNING

To evaluate the precision of our noisy sample detection, we conducted a comparison between our predictions and pixel-wise annotations of the Kubric dataset. Table 2 presents our method's performance, which includes SI and SI+Seg. Both configurations exhibit significantly high performance, with our full framework (SI+Seg.) reaching a maximum accuracy of up to 98%.

### 4.5 ABLATION STUDY

To assess the sensitivity of our influence score thresholding, we conducted an ablation study. As shown in Table 3, finding the appropriate sweet spot without a validation set can be challenging. Therefore, we incorporated the segmentation method to somewhat alleviate this issue.

## 5 RELATED WORK

**Robust Learning in NeRF** Recent work addresses the limitation of NeRF, which previously required well-curated training data. Thanks to these efforts, NeRF has become robust to various noise sources, including inaccurate camera poses (Lin et al., 2021; Bian et al., 2023), motion blur (Ma et al., 2022), and the presence of transient objects (Marí et al., 2022; Sabour et al., 2023). Specifically, several approaches handle situations where the 3D consistency among training images is disrupted. A common approach to enabling NeRF to adapt to appearance variations between views is to assign learnable codes to each training view (Martin-Brualla et al., 2021; Chen et al., 2022). This approach has proven to be robust in the presence of appearance variations such as lighting. In cases where training images contain transient objects, one can use a segmentation model (Tancik et al., 2022) to mask out transient objects, while another approach jointly trains the uncertainty of each pixel, with high uncertainty pixels receiving reduced weight in the loss (Marí et al., 2022; Martin-Brualla et al., 2021). RobustNeRF (Sabour et al., 2023) assumes that transient objects occupy large and connected regions, and employs trimmed least squares to train NeRF. However, these methods may rely on strong assumptions or introduce artifacts in the results. NeRF designed for dynamic scenes (Wu et al., 2022; Gao et al., 2021; Li et al., 2021; Xian et al., 2021) can also separate the static component within the scene. However, this approach requires input data to be provided in a video-like format for optimal results.

**Influence Function** Influence Function (Hampel, 1974; Koh & Liang, 2017)) and its approximations (Pruthi et al., 2020; Schioppa et al., 2022) have been utilized in various deep learning tasks by estimating the influence of a data point on the model's prediction. Also, self-influence is often used when detecting minority samples, such as mislabeled ones, without a validation set (**?**).

**Segmentation** Segmentation is an essential component in many visual understanding systems, which plays a vital role in various applications including medical image analysis (Ronneberger et al., 2015; Milletari et al., 2016; Çiçek et al., 2016; Hatamizadeh et al., 2022; 2021), autonomous

vehicles (Siam et al., 2017; Feng et al., 2020; Zendel et al., 2022), and point cloud analysis (Qi et al., 2017a;b; Zhao et al., 2021) to count a few. Recently, some approaches (Bucher et al., 2019; Ding et al., 2022; Kirillov et al., 2023; Zou et al., 2023) have examined zero-shot segmentation, extending segmentation models to handle novel object classes at test time. Among these studies, SAM (Kirillov et al., 2023) has garnered significant attention for its outstanding zero-shot segmentation performance across various datasets for panoptic segmentation. Additionally, SAM supports prompt input by the user, enabling interactive segmentation.

## 6  CONCLUSION

In this paper, we proposed a noise pruning framework leveraging influence functions with segmentation-based refinement. We first pioneered the integration of influence functions into NeRF, effectively pruning distractors that disrupt the 3D consistency. Also, we incorporated panoptic segmentation with influence scores that enhance the precision of pixel-level noise candidates. We believe that our dataset pruning approach brining out the potential of easier data construction in NeRF.

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
