# OpenReview forum: "Object-Centric Noise Filtering in Neural Radiance Fields via Influence Functions and Segmentation"
_ICLR.cc/2024/Conference — ICLR 2024 Conference Withdrawn Submission_

### Official Review · Reviewer_hbFT · 2023-10-29

**Soundness:** 2 fair
**Presentation:** 3 good
**Contribution:** 2 fair
**Rating:** 5
**Confidence:** 3

**Summary:**

The paper presents a method to identify distractors during the NeRF reconstruction. With the help of SAM segmentation, it can effectively group the segments belonging to the same objects while ignoring the distractors. Experimental results shown that the proposed method is effective in improving the NeRF reconstruction quality.

**Strengths:**

The proposed method is simple, and it demonstrated better results on multiple NeRF dataset.

**Weaknesses:**

The proposed method is highly incremental and it is a straight forward extension of RobustNeRF with SAM segmentation. There are not much analyses why there are distractors and how sensitive is the proposed method against the segmentation accuracy. It basically assume the SAM segmentation is good enough such that the segments belong to the same object can be grouped successfully using the proposed method in Sec. 3. Although it demonstrate some good results, the scene are relatively simple which has only one dominant object. I am not fully convinced that the method proposed in Sec. 3 is general enough for handling complex scene with many objects. I am also not convinced that the proposed method can handle tiny objects effectively since the tiny objects are likely to be filtered out by the proposed method as distractors.

Considering the limited technical novelty and the potential problems on complex scene and tiny objects, I am not convinced that the submission has reach the bar of acceptance.

**Questions:**

Please provide additional experiments or examples to convince that the proposed method is effective in handling complex scenes and tiny objects.

---

### Official Review · Reviewer_RkM8 · 2023-10-30

**Soundness:** 2 fair
**Presentation:** 1 poor
**Contribution:** 2 fair
**Rating:** 3
**Confidence:** 4

**Summary:**

This paper proposes a noise pruning pipeline for NeRF learning, aimed at training NeRF on images with distortion. Specifically, it proposes to first use inference functions to determine the potential pixels that are considered as distortion. Then, it segments out regions that include pixels considered as distortion using the SAM model. Finally, it removes the segmented-out pixels from the training images to reduce the inference of distortion. The paper then conducts experiments on the RobustNeRF dataset and a synthetic dataset.

**Strengths:**

1. Learning NeRF in a robust manner is important and useful.
2. Using the inference function to determine the distortion seems reasonable.

**Weaknesses:**

1. I believe the proposed pipeline may have limitations in real-world applications. In real-world scenarios, noise and distortion can originate from various sources, such as inaccurate camera calibration, motion blur, out-of-focus blur, JPEG compression noise, and more. Simply discarding the information from these inconsistent pixels by segmenting out the entire region may result in significant information loss, especially in cases like inaccurate camera calibration and motion blur. I would recommend that the author reconsider this approach and seek to improve it.
2. The presentation of the results is limited. It is challenging to assess the effectiveness of the approach based on the content in the current version of the paper. I suggest that the author include more qualitative results, as there is more than half a page remaining empty.
3. The Method section is not well-written, containing numerous typos and mistakes. For example, the line following equation 5 states "loss is defined as," but it proceeds to show how theta is computed. Additionally, the notion of I(r, r) appears to be unscientific.

**Questions:**

1. How does the SAM is actually prompted? Is there a non-maximum suppression process used on top of the selected pixels?
2. What will happen if this approach is applied to clean images without any noise? Since it seems a fixed number of pixels are considered as noise in stage 1.
3. What if this approach is applied to JPEG compression noise?

**Details Of Ethics Concerns:**

No concerns.

---

### Official Review · Reviewer_yQMB · 2023-10-31

**Soundness:** 2 fair
**Presentation:** 2 fair
**Contribution:** 1 poor
**Rating:** 3
**Confidence:** 4

**Summary:**

As there would be dynamic distractors in the scene which usually occlude the objects-of-interest, NeRF models could not reconstruct the 3D scene correctly. This paper adopts Influence Functions to evaluate each pixel in the training set and finds out the distractor pixels. To further improve the consistency in each image plane, it also integrates a segmentation method, i.e., SAM. The experimental results verify the feasibility of the proposed method.

**Strengths:**

This paper firstly adopts Influence Functions to find the distractors in the data for NeRF reconstruction, which provides a new method to alleviate this difficulty.

The paper is easy to understand and the presentation is acceptable.

**Weaknesses:**

-- The novelties are limited. As the key idea of this paper is adopting an existing method, i.e., Influence Functions, to improve the robustness of NeRF reconstruction, it is acceptable if the performance gain is remarkable. Unfortunately, this is not true according to the experiments.

-- There are many confusing results and settings in the experiments, which significantly reduce their credibility. In Tab. 1, there are no results of RobustNeRF. Acctually, according to the RobustNeRF paper, it achieves better performance overall. In Tab. 4, the results of RobustNeRF are not consistent with the results reported by the RobustNeRF paper. Moreover, the results of Crab scene are missed.

**Questions:**

What are the principles of the experimental settings? Why are some results missed or inconsistent?

This paper has not reported the computation cost of the method. It seems really expensive to compute the Hessian matrix.

I have not found the appendix.

**Details Of Ethics Concerns:**

None.